

# Enhancing fruit freshness classification with adaptive knowledge distillation and global response normalization in convolutional networks

Semih Demirel[1] and Oktay Yıldız[2]

[1] Department of Information Systems, Gazi University Ankara, Ankara, Turkey
[2] Department of Computer Engineering, Gazi University Ankara, Ankara, Turkey

## ABSTRACT

The assessment of fruit freshness is crucial for ensuring food quality and reducing waste in agricultural production. In this study, we propose *Global Response Normalization and Gaussian Error Linear Unit Enhanced Network (GGENet)*, a novel deep learning architecture that leverages adaptive knowledge distillation (AKD) and global response normalization (GRN) to classify fruits as fresh or rotten. Our model comprises two variants: *GGENet-Teacher (GGENet-T)*, serving as the teacher model, and *GGENet-Student (GGENet-S)*, functioning as the student model. By transferring attention maps from the teacher to the student model, we achieve efficient adaptive knowledge distillation, enhancing the performance of the lighter student model. Experimental results demonstrate that the *GGENet with adaptive knowledge distillation (GGENet-AKD)* achieves a competitive accuracy of 0.9818, an F1-score of 0.9818, and an area under the curve (AUC) score of 0.9891. The proposed method significantly contributes to reducing food waste and enhancing quality control in agriculture by facilitating early detection of rotting fruits.

## INTRODUCTION

Food and agriculture play a major role in human society (*Choithani, Jaleel & Rajan, 2024*). The food crisis has emerged as one of humanity's biggest issues, along with climate change (*Burkhanov et al., 2024*). In addition to this food problem, there is a large amount of food waste. Foods that decay and deteriorate become unsuitable for consumption, especially during the production stage (*Sultana, Jahan & Uddin, 2022*).

The same problems can also be seen in fruits, which are an important food product for humans grown in the field. Especially in the production stage, rotting fruits are set aside without being used (*Gupta & Tripathi, 2024*). This situation further exacerbates the already existing food issue.

Processes for quality control are essential for reducing food waste. Many technologies are employed, particularly in the production of fruits and vegetables, to preserve the freshness and usability of harvested goods (*Nath et al., 2024*). Artificial intelligence

Corresponding author
Semih Demirel,
semihhdemirel96@gmail.com

algorithms and image processing techniques are commonly employed in modern agriculture to assess the quality of harvested goods and identify potentially spoiling commodities early on *Pandey & Mishra (2024)*.

Waste can be avoided by quickly identifying and separating products that are susceptible to spoiling. Reducing food waste through such strategies is crucial for environmental sustainability as well as economic viability (*Burkhanov et al., 2024*). Thus, reducing food waste guarantees a more effective use of natural resources in addition to improving food security.

Technology, especially developments in computer vision and deep learning, provides effective methods for maintaining and monitoring food quality, allowing for the large-scale resolution of this problem. Real-time sorting process optimization, produce classification, and automatic early spoiling detection are all possible with these technologies.

Convolutional neural networks (CNNs), which analyze visual information by learning spatial hierarchies through layers of convolutional filters, are among the most efficient designs used in deep learning for image processing. Convolutional kernels are used by convolutional neural networks to extract hierarchical characteristics from images at various sizes (*Xiao et al., 2024*). To add non-linearity to the network, an activation function is added after the convolution process. All negative pixel values in the feature map are set to zero by the rectified linear unit (ReLU) (*Rajesh Kanna & Kumararaja, 2024*).

The rectified linear unit and other activation functions have been outperformed by the Gaussian error linear unit (GELU) (*Hendrycks & Gimpel, 2016*). The model becomes non-linear when using either the rectified linear unit or the Gaussian error linear unit (*Wu, Du & Heng, 2024*). Training can make more subtle alterations because of its smoothness, which may enhance learning outcomes. The Gaussian error linear unit allows some negative values to pass through, but the rectified linear unit entirely filters negative inputs (*Khan et al., 2024*).

Channel and spatial attention are used to enhance feature representation by focusing on the most important aspects of an input feature map (*Qin et al., 2024*). Channel attention focuses on emphasizing certain channels in a feature map, while spatial attention highlights specific spatial locations (*Liu, Qian & Wang, 2024*). Convolutional block attention module, developed by *Woo et al. (2018)*, applies spatial and channel attention sequentially. Because of this combination, the model is able to focus on the spatial areas and channels that need the most attention.

Normalization helps models train faster by standardizing inputs or network activations (*Liu et al., 2024*). Stabilizing responses within a network, particularly in convolutional layers, is the aim of the normalization technique called global response normalization (GRN) (*Woo et al., 2023*). Global response normalization works by bringing the feature responses into uniformity across all spatial dimensions. Batch normalization (*Ioffe & Szegedy, 2015*) stabilizes and accelerates training while preserving the integrity of the input distribution by leveling the inputs to each layer.

A smaller model is trained to simulate the actions of a larger model in knowledge distillation (*Hinton, 2015*). Instead of training the smaller model directly on the hard labels, the larger model's output probabilities are used (*Ding et al., 2024*). To produce the

probabilities, the output logits of the larger model are divided by a temperature parameter before applying the softmax function. The weights of the loss function can be dynamically adjusted during training. In epochs where the smaller model performs well, higher weight is given to increase the influence of the smaller model (*Ma et al., 2024*).

In this study, for the purpose of classifying rotten and fresh fruits, we developed a new model called *Global Response Normalization and Gaussian Error Linear Unit Enhanced Network (GGENet)*, designed in two variants: *GGENet-Teacher (GGENet-T)* as the teacher and *GGENet-Student (GGENet-S)* as the student. The teacher model's attention maps were transferred to the student model by adaptive knowledge distillation (AKD), which improved the student's ability to acquire attention knowledge. The dynamic weighting strategy employed during training allows the student model to progressively rely more on its learned representations. The student model, to which adaptive knowledge distillation was applied, is named *GGENet with adaptive knowledge distillation (GGENet-AKD)*. The proposed approach has demonstrated that *GGENet-S*, with guidance from *GGENet-T*, achieves competitive performance in classifying rotten and fresh fruits.

The necessity for a quick and high-performing deep learning model for classifying fruit freshness in practical settings is the primary motivation for this investigation. We selected the fruit freshness classification challenge to assess our adaptive knowledge distillation method due to its technical and practical importance. Food waste is a significant societal issue that is addressed by providing real-time identification of bad products. The objective of creating lightweight, effective models is also in line with its implementation on low-power devices. It is also a trustworthy and repeatable benchmark because well-annotated public datasets are readily available.

In contrast to previous studies that focused solely on improving network accuracy or reducing model size, our approach balances both objectives by introducing a scalable and efficient architecture for evaluating fruit quality. Although previous research examined knowledge distillation or attention mechanisms separately, our approach combined spatial and channel attention in a novel way with a lightweight design. Moreover, adaptive knowledge distillation was used to transfer attention-specific knowledge from a teacher model. Furthermore, the addition of the global response normalization-gaussian error linear unit convolution (GRN-GELU Conv) distinguishes our model by enhancing feature representation *via* stabilized normalization and smooth activation. With this method, the student model can perform competitively with a much lower computational cost, which makes it appropriate for resource-constrained, real-world agricultural applications.

The following are the primary contributions of our work:

- We developed a new *GGENet* model that efficiently transfers the attention knowledge learned from the teacher to the student model, ensuring both performance and inference time efficiency.
- A new module named GRN-GELU Conv module was designed. In this module, a residual structure (*He et al., 2016*) was used along with global response normalization and Gaussian error linear unit activation to enhance feature extraction in convolutional networks.

- The network integrates the channel and spatial attention at multiple stages, enhancing the network's ability to focus on relevant spatial and channel wise features.
- The model utilizes a hierarchical downsampling approach through the custom downsample layers, progressively reducing the spatial dimensions while increasing the channel depth.
- By combining global response normalization, Gaussian error linear unit activation, and attention modules, the network systematically aggregates features across different scales and layers.

'Related Works' examines relevant research in the domains of knowledge distillation, attention mechanisms, and fruit classification. 'Materials and Method' contains the attention modules, the dataset, the suggested *GGENet* architecture, and the adaptive knowledge distillation technique. The experimental setting, evaluation metrics, and outcomes proving the efficacy of our strategy are described in 'Experimental Results'. The article's conclusions, key findings, and potential directions for future research are presented in 'Conclusion'.

## RELATED WORKS

A study by *Fahad et al. (2022)* divided fruits and vegetables into three distinct categories: decaying, pure-fresh, and medium-fresh, using an inventive automated system. The proposed system utilized two deep learning models: You Only Look Once (YOLO) (*Redmon et al., 2016*) and Visual Geometry Group 16 (VGG16) (*Simonyan & Zisserman, 2014*). Whereas the YOLO model locates objects in the images, VGG16 focuses on categorizing the objects and assessing their freshness. According to experimental evaluations, the technique achieved a mean average precision (mAP) of 0.84 and accuracies of 0.82 and 0.84 with VGG16 and YOLO, respectively.

*Arivalagan et al. (2024)* employed a dataset in their study that includes fruits such as guavas, apples, and bananas, captured under various lighting and angle conditions. For each fruit species, the dataset was utilized to complete six binary classification tasks that were responsible detecting between fresh and rotting samples. The quality assessment was carried out by processing the dataset and analyzing its features using neural networks. With an accuracy rate of 0.996, Inceptionv3 (*Szegedy et al., 2016*) emerged as the most accurate model among the neural network architectures.

*Dakwala et al. (2022)* used pretrained convolutional neural network architectures to automatically classify products into fresh and harmful categories. To determine which of the proposed models was most appropriate for real-time supply chain applications, its speed and accuracy were evaluated. A Kaggle dataset containing images of oranges, bananas, and both fresh and rotten apples was used in the study. Data augmentation methods like rescaling and random flipping were used to enhance model generalization.

Using cutting-edge deep learning models, *Akshi et al. (2024)* presented an accurate and effective approach for identifying and classifying fruits and vegetables based on their freshness. The deep learning models, YOLO and VGG16, were used to examine key characteristics indicative of freshness, such as color, texture, shape, and size. The study

demonstrated the effectiveness of the automated system on a large and diverse dataset of fruits and vegetables under various conditions.

Finding the algorithm that best predicts the freshness of fruits based on user-provided images was the primary objective of the study by *Rohit Mamidi et al. (2022)*. The study was split into two parts: the first looked at how machine learning is used in daily life, and the second assessed how well different models performed on a particular dataset. Images of both fresh and rotting fruits were used for binary and multi-class classification tasks. Deep learning models were compared with a number of machine learning approaches. The outcomes showed that deep learning methods performed better than conventional machine learning models.

In the research by *Sia & Baco (2023)*, the optimal configurations for properly determining the freshness of oranges, apples, and bananas were determined by adjusting the hyperparameters of a convolutional neural network model. To assess the model with different combinations of these parameters, the rectified linear unit was utilized. The convolutional neural network model with hyperparameter tuning achieved the highest classification accuracy of 0.9904, as shown by the experimental results.

*Sangeetha et al. (2024)* presented a thorough methodology in their investigation, which starts with gathering a dataset of both fresh and decaying fruits. The fundamental component of the method is the use of convolutional neural networks to classify fruits into different groups. To distinguish between rotten and fresh fruits, the convolutional neural network model was trained on a wide range of images. In addition, the project incorporated open computer vision library to evaluate fruit maturity according to color.

*Arun Kumar et al. (2023)* focused on finding the best deep learning algorithms for determining fruit freshness and quality. In particular, the study used banana fruits and applied image augmentation to boost the dataset's image count. In order to improve prediction accuracy, the study combined residual networks with support vector machines. The study's findings demonstrated the high efficiency of the proposed ensemble machine learning technique, which achieved an accuracy rate of 0.9912 on the expanded dataset and 0.9878 on the original dataset.

Our research offers multiple significant insights that set it apart from other studies in the field. We provide a *GGENet* model that can effectively transfer the teacher's attention knowledge to the student model. In comparison to existing techniques, this unique approach aims to offer a lighter and higher-performing model. Furthermore, our research leverages the benefits of the Gaussian error linear unit structure and global response normalization to improve the results of the student model and encourage deeper learning through the efficient integration of the residual structure, Gaussian error linear unit, and global response normalization into the convolutional layers.

## MATERIALS AND METHOD

### Dataset

This project aims to classify six distinct fruits as fresh or rotting in order to boost agricultural output. We used an open source dataset that we downloaded from the Kaggle website for this purpose. Table 1 contains information about the dataset.

| Table 1 Dataset for fruit classification. | |
|---|---|
| **Classes** | **Number of images** |
| Fresh apples | 2,088 |
| Fresh bananas | 1,962 |
| Fresh oranges | 1,854 |
| Rotten apples | 2,943 |
| Rotten bananas | 2,754 |
| Rotten oranges | 1,998 |
| Total | 13,599 |

Five folds were created from the dataset, and training was done for each fold. The dataset was split into 80% training and 20% testing data for each fold. The validation set was then created within each fold using 10% of the training data.

## Global response normalization

Ensuring that the data has a consistent scale, minimizing internal covariate shift, accelerating convergence, and enhancing generalization are the main objectives of normalization.

Global response normalization works by normalizing the activations of a layer across the entire feature map rather than just within each feature map (*Woo et al., 2023*). This means that instead of normalizing each feature map individually, global response normalization normalizes the responses of all feature maps collectively. Unlike batch normalization, which normalizes activations within a batch, global response normalization normalizes activations across the entire feature map or network, potentially making it more suited for tasks where batch statistics are not as effective (*Woo et al., 2023*).

In Eq. (1), $G_x$ represents the global response of the input tensor $x \in \mathbb{R}^{H \times W \times C}$.

$$G_x = \sqrt{\sum_{i,j} x_{i,j}^2}. \tag{1}$$

The given equation is calculated using euclidean normalization over the spatial dimensions $i$ and $j$. This normalization process is detailed in Eq. (2).

$$N_x = \frac{G_x}{\text{mean}(G_x) + 10^{-6}} \tag{2}$$

where $N_x$ is the normalized global response. $G_x$ is normalized by dividing it by the mean of $G_x$ across the channels, with an addition of $10^{-6}$ to prevent division by zero. The final output is given in Eq. (3).

$$y = \gamma \cdot (x \cdot N_x) + \beta + x \tag{3}$$

where $y$ is the final output of the global response normalization layer, and $\gamma$ and $\beta$ are learnable parameters. The algorithmic representation of the global response normalization layer is provided in Algorithm 1.

---

**Algorithm 1**   **Global response normalization layer.**

**Require:** Input tensor $x$ with dimensions *(N, C, H, W)*

1: Initialize learnable parameters $\gamma$ and $\beta$ with shape $(1, 1, 1, C)$
2: Compute global response $G_x \leftarrow \sqrt{\sum_{i,j} x_{i,j}^2}$ over spatial dimensions *(H, W)*
3: Normalize $N_x \leftarrow \frac{G_x}{\text{mean}(G_x) + 10^{-6}}$
4: Compute output $y \leftarrow \gamma \cdot (x \cdot N_x) + \beta + x$
5: **return** $y$

---

## Attention module

Convolutional neural networks can benefit from the incorporation of the lightweight convolutional block attention module, which improves feature representation (*Li et al., 2021*). The input feature maps go through to channel attention and spatial attention in turn. Spatial attention chooses which area of each feature map to concentrate on, whereas channel attention chooses which feature map to emphasize (*Wang et al., 2021*). Equations (4), (5), (6), and (7) provide the mathematical operations for channel attention, while Equations (8), (9), and (10) provide the mathematical operations for spatial attention.

Given an input tensor $x \in \mathbb{R}^{C \times H \times W}$, where $C$ is the number of channels, $H$ is the height, and $W$ is the width, global average pooling (GAP) and global max pooling (GMP) are applied to the input tensor $x$ in Eq. (4).

$$x_{max} = GMP(x) \in \mathbb{R}^{C \times 1 \times 1} \quad x_{avg} = GAP(x) \in \mathbb{R}^{C \times 1 \times 1} \tag{4}$$

where GAP and GMP represent global average pooling and global max pooling, respectively. Equation (5) provides the outputs of the fully connected layers applied to $x_{max}$ and $x_{avg}$.

$$z_{max} = FC_2(ReLU(FC_1(x_{max}))) \in \mathbb{R}^{C \times 1 \times 1} \quad z_{avg} = FC_2(ReLU(FC_1(x_{avg}))) \in \mathbb{R}^{C \times 1 \times 1} \tag{5}$$

where $FC$ and $ReLU$ represent the fully connected layer and rectified linear unit, respectively. The sum of $z_{max}$ and $z_{avg}$ is given in Eq. (6).

$$z = z_{max} + z_{avg} \in \mathbb{R}^{C \times 1 \times 1}. \tag{6}$$

The output of the channel attention is given in Eq. (7).

$$out_c = x \odot \sigma(z) \tag{7}$$

where $out_c$ and $\sigma$ represent the output of channel attention and the sigmoid function, respectively. The symbol $\odot$ denotes element-wise multiplication. Figure 1 shows the architecture of channel attention.

The input tensor $x$ undergoes exposure to maximum and average pooling in spatial attention. The formulas for the average and maximum pooling procedures are given in Eq. (8).

$$x_{max} = MaxPool(x) \in \mathbb{R}^{1 \times H \times W} \quad x_{avg} = AvgPool(x) \in \mathbb{R}^{1 \times H \times W} \tag{8}$$

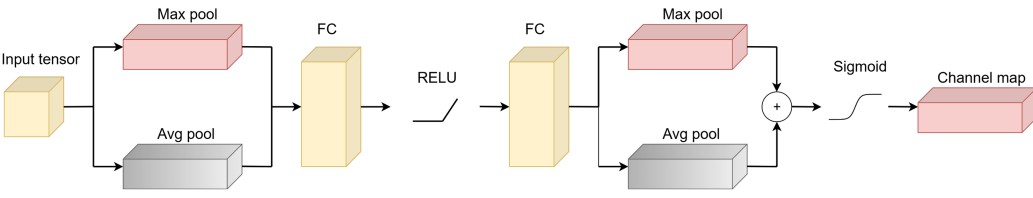

**Figure 1 Architecture of channel attention.**

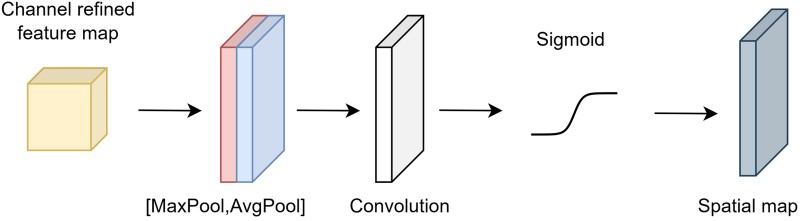

**Figure 2 Architecture of spatial attention.**

where *MaxPool* and *AvgPool* represent the maximum and average pooling, respectively. The concatenation of $x_{max}$ and $x_{avg}$ is given in Eq. (9).

$$x_{concat} = concat([x_{avg}, x_{max}], dim = 1) \in \mathbb{R}^{2 \times H \times W} \qquad (9)$$

where $x_{concat}$ represents the concatenation of the average and maximum pooling of the input tensor $x$. The output of the spatial attention mechanism is given in Eq. (10).

$$out_s = x \odot \sigma(conv(x_{concat}) \in \mathbb{R}^{1 \times H \times W}) \qquad (10)$$

where $out_s$ is the output of the spatial attention, *conv* represents the convolutional layer, and $\odot$ denotes element-wise multiplication.

Figure 2 shows the architecture of spatial attention.

The output of the channel and spatial attention applied sequentially is given in Eq. (11).

$$out\,put_{final} = out_s \odot (out_c \odot x) \qquad (11)$$

where $out\,put_{final}$ is obtained by applying element-wise multiplication to the outputs of channel and spatial attention sequentially.

Figure 3 illustrates the final output of attention module.

## Adaptive knowledge distillation

The student model in the conventional knowledge distillation receives training information from the teacher model (*Mi, Wermter & Zhang, 2024*). Through the temperature parameter, the hard labels of the teacher model are converted into soft labels (*Zhang et al., 2024*). The student model is trained using both hard labels and soft labels. A specific weight value is assigned to each target label. In the traditional knowledge distillation method, these weights are fixed. The total loss is calculated as given in Eq. (12).

$$loss_{total} = (1 - \alpha).loss_{CE} + \alpha.loss_{distill} \qquad (12)$$

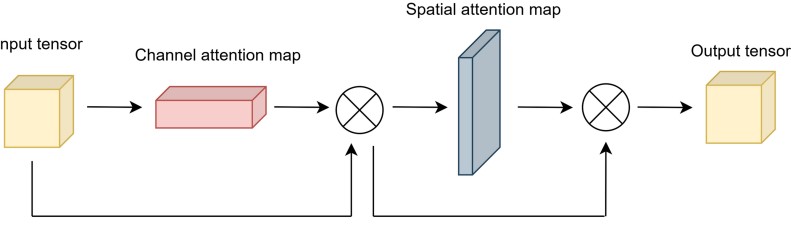

**Figure 3 Architecture of attention module.**

where $loss_{CE}$ and $loss_{distill}$ represent the cross-entropy and distillation loss, respectively. The weight parameter $\alpha$ is constant in knowledge distillation.

A more modern approach, known as adaptive knowledge distillation, uses a dynamic weight parameter. The weight parameter increases or decreases depending on the teacher model's loss value in each epoch (*An et al., 2024*). If the teacher model's loss value is high, the weight for the teacher model is kept lower. Conversely, if the loss value is low, a higher weight is assigned, giving the teacher model more influence on the total loss. Dynamic $\alpha$ parameter is calculated as given in Eq. (13).

$$\alpha = \frac{loss_{CE}}{loss_{CE} + loss_{distill}} \tag{13}$$

where $loss_{CE}$ and $loss_{distill}$ are the loss values obtained from the student and teacher models during training, respectively. An increase in $loss_{distill}$ causes the $\alpha$ value to decrease, thereby reducing the impact of the knowledge transferred from the teacher model on the total loss.

The training process is strengthened and made more effective by this adaptive mechanism, which allows the student model to selectively rely on the teacher's instructions based on its dependability. When the teacher is ambiguous, the model prevents overfitting to possibly incorrect soft targets by dynamically modifying the teacher's effect, particularly during the early training phases or on difficult datasets. Algorithm 2 presents the algorithmic representation of the adaptive knowledge distillation.

The adaptive weight parameter $\alpha$, which is a crucial component of the method, is determined by dividing $loss_{CE}$ by the sum of $loss_{CE}$ and $loss_{distill}$. How big of an impact each loss should have on the overall loss is determined by this weight. The method lessens the influence of the teacher's direction by decreasing $\alpha$.

## Proposed model

In this study, we designed a new network for rotten and fresh fruit classification. The model we developed is a classification model consisting of two separate networks. There are teacher and student models named *GGENet-T* and *GGENet-S*, each with different numbers of parameters. *GGENet-T* consists of approximately 7.1 million parameters, while *GGENet-S* consists of approximately 2.3 million parameters. The objective is to use adaptive knowledge distillation to move the attention knowledge from the teacher's model to the attention map of the student model. An illustration of the proposed network is presented in Fig. 4.

**Algorithm 2** **Adaptive knowledge distillation.**

**Require:** Teacher model $T$, student model $S$, temperature $\tau$

1: Initialize student model parameters

2: **for** each training epoch **do**

3:     **for** each mini-batch $(x, y)$ **do**

4:         Compute teacher output: $p_T \leftarrow \mathrm{softmax}(T(x)/\tau)$

5:         Compute student output: $p_S \leftarrow \mathrm{softmax}(S(x)/\tau)$

6:         Compute hard label loss: $loss_{CE} \leftarrow \mathscr{L}_{CE}(S(x), y)$

7:         Compute distillation loss: $loss_{distill} \leftarrow \mathscr{L}_{KL}(p_S, p_T)$

8:         Compute adaptive weight: $\alpha \leftarrow \frac{loss_{CE}}{loss_{CE}+loss_{distill}}$

9:         Compute total loss: $loss_{total} \leftarrow (1-\alpha) \cdot loss_{CE} + \alpha \cdot loss_{distill}$

10:        Update student model parameters using gradient descent on $loss_{total}$

11:     **end for**

12: **end for**

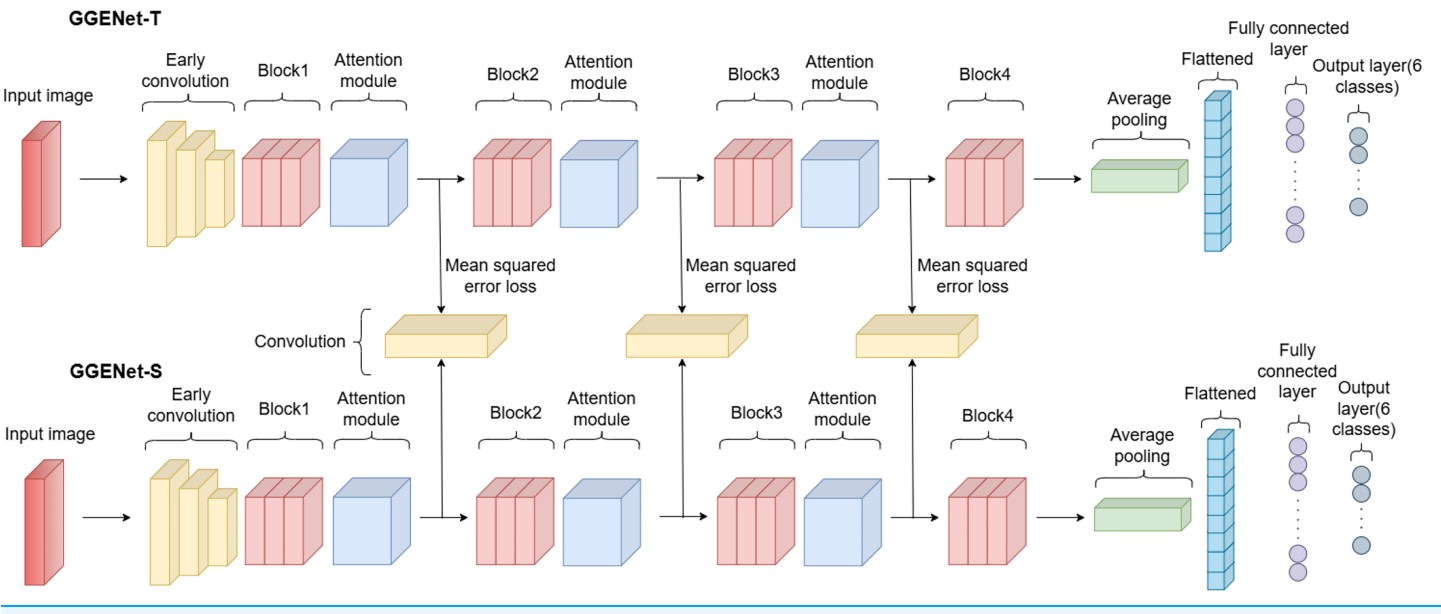

**Figure 4 Architecture of proposed network.**

The number of layers in the student and teacher models is the same, but the number of parameters varies. The reason for the higher parameter count in the teacher model is the use of double the number of channels in the convolutional layers. The table representations of the *GGENet-T* and *GGENet-S* models are provided in Tables 2 and 3, respectively.

When both tables are examined, it is clear that the network structure is identical for both models. All that differs, though, is the amount of input and output channels. As the number of channels decreases, the number of parameters also decreases. An early convolution is applied to the input image in the models. Following this convolution, global response normalization and the Gaussian error linear unit are used to normalize the

**Table 2 Table representation of *GGENet-T*.**

| Component | Layer | Input | Output | Shape |
|---|---|---|---|---|
| Early convolution | Convolution (k = 3, s = 2, p = 1) | 3 | 32 | 112 × 112 |
| | Global response normalization | 32 | 32 | 112 × 112 |
| | Gaussian error linear unit | 32 | 32 | 112 × 112 |
| Block1 | GRN-GELU Conv module | 32 | 32 | 112 × 112 |
| | GRN-GELU Conv module | 32 | 32 | 112 × 112 |
| | Downsample layer | 32 | 64 | 56 × 56 |
| Attention module | Convolutional block attention module | 64 | 64 | 56 × 56 |
| Block2 | GRN-GELU Conv module | 64 | 64 | 56 × 56 |
| | GRN-GELU Conv module | 64 | 64 | 56 × 56 |
| | Downsample layer | 64 | 128 | 28 × 28 |
| | GRN-GELU Conv module | 128 | 128 | 28 × 28 |
| | GRN-GELU Conv module | 128 | 128 | 28 × 28 |
| | Downsample layer | 128 | 256 | 14 × 14 |
| Attention module | Convolutional block attention module | 256 | 256 | 14 × 14 |
| Block3 | GRN-GELU Conv module | 256 | 256 | 14 × 14 |
| | GRN-GELU Conv module | 256 | 256 | 14 × 14 |
| | GRN-GELU Conv module | 256 | 256 | 14 × 14 |
| | Downsample layer | 256 | 512 | 7 × 7 |
| | GRN-GELU Conv module | 512 | 512 | 7 × 7 |
| | GRN-GELU Conv module | 512 | 512 | 7 × 7 |
| | GRN-GELU Conv module | 512 | 512 | 7 × 7 |
| | Downsample layer | 512 | 1,024 | 3 × 3 |
| Attention module | Convolutional block attention module | 1,024 | 1,024 | 3 × 3 |
| Block4 | GRN-GELU Conv module | 1,024 | 1,024 | 3 × 3 |
| | GRN-GELU Conv module | 1,024 | 1,024 | 3 × 3 |
| | GRN-GELU Conv module | 1,024 | 1,024 | 3 × 3 |
| Classifier | Adaptive average pooling | 1,024 | 1,024 | 1 × 1 |
| | Linear | 1,024 | 6 | |

activations and provide a non-linear transformation. Global response normalization, in particular, balances the overall activation level of each feature map, enhancing the model's learning capacity and reducing overfitting (*Woo et al., 2023*). Subsequently, the Gaussian error linear unit activation, compared to other activation functions like the rectified linear unit, provides a smoother non-linearity, contributing to more stable and effective learning by the model (*Hendrycks & Gimpel, 2016*).

As the attention module, the convolutional block attention module is employed with the same structure as depicted in Fig. 3. In this module, channel attention is applied first, followed by spatial attention in sequence.

**Table 3** Table representation of *GGENet-S*.

| Component | Layer | Input | Output | Shape |
|---|---|---|---|---|
| Early convolution | Convolution (k = 3, s = 2, p = 1) | 3 | 16 | 112 × 112 |
| | Global response normalization | 16 | 16 | 112 × 112 |
| | Gaussian error linear unit | 16 | 16 | 112 × 112 |
| Block1 | GRN-GELU Conv module | 16 | 16 | 112 × 112 |
| | GRN-GELU Conv module | 16 | 16 | 112 × 112 |
| | Downsample layer | 16 | 32 | 56 × 56 |
| Attention module | Convolutional block attention module | 32 | 32 | 56 × 56 |
| Block2 | GRN-GELU Conv module | 32 | 32 | 56 × 56 |
| | GRN-GELU Conv module | 32 | 32 | 56 × 56 |
| | Downsample layer | 32 | 64 | 28 × 28 |
| | GRN-GELU Conv module | 64 | 64 | 28 × 28 |
| | GRN-GELU Conv module | 64 | 64 | 28 × 28 |
| | Downsample layer | 64 | 128 | 14 × 14 |
| Attention module | Convolutional block attention module | 256 | 256 | 14 × 14 |
| Block3 | GRN-GELU Conv module | 128 | 128 | 14 × 14 |
| | GRN-GELU Conv module | 128 | 128 | 14 × 14 |
| | GRN-GELU Conv module | 128 | 128 | 14 × 14 |
| | Downsample layer | 128 | 256 | 7 × 7 |
| | GRN-GELU Conv module | 256 | 256 | 7 × 7 |
| | GRN-GELU Conv module | 256 | 256 | 7 × 7 |
| | GRN-GELU Conv module | 256 | 256 | 7 × 7 |
| | Downsample layer | 256 | 512 | 3 × 3 |
| Attention module | Convolutional block attention module | 512 | 512 | 3 × 3 |
| Block4 | GRN-GELU Conv module | 512 | 512 | 3 × 3 |
| | GRN-GELU Conv module | 512 | 512 | 3 × 3 |
| | GRN-GELU Conv module | 512 | 512 | 3 × 3 |
| Classifier | Adaptive average pooling | 512 | 512 | 1 × 1 |
| | Linear | 512 | 6 | |

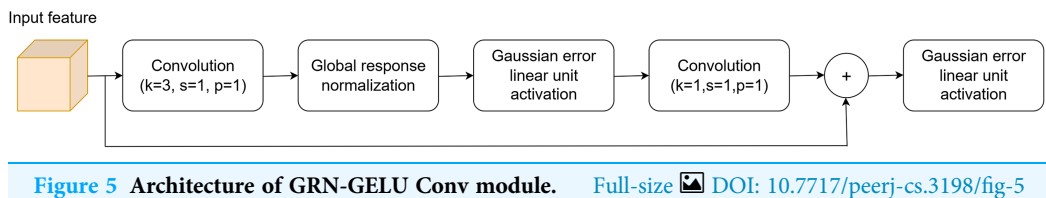

**Figure 5** **Architecture of GRN-GELU Conv module.**

### GRN-GELU Conv module

Within the scope of this network, a new GRN-GELU Conv module was designed. The designed module is utilized in both networks. The architecture of the GRN-GELU Conv module is shown in Fig. 5.

In the GRN-GELU Conv module, a convolution is applied using a 3 × 3 kernel size as used in the study by *Howard (2017)*. Global response normalization is applied after the

convolutional layer with the $3 \times 3$ kernel to stabilize the network and ensure that the output of the convolutional layers is normalized in a way that accounts for the entire spatial structure. After the global response normalization, Gaussian error linear unit activation is applied, which allows for smooth non-linearity. A second convolution is applied with a kernel size of 1 (*Howard, 2017*). Through a residual connection, the input feature is added to the output of the second convolution. After the residual connection, Gaussian error linear unit activation is applied again to ensure smooth non-linearity.

The differences between our module and the module proposed by *Woo et al. (2023)* are that we did not use the third convolution layer with a kernel size of 1. Additionally, after the first convolution layer, we used the GRN-GELU structure instead of layer normalization. After the residual connection, we applied the Gaussian error linear unit activation.

After the GRN-GELU Conv module, a downsample layer is applied. The primary goal of the downsample layer is to reduce the spatial resolution of the input feature map by a factor of 2 using a $2 \times 2$ convolutional kernel with a stride of 2. Additionally, the downsample layer doubles the number of channels.

### Distilling the attention knowledge

The main purpose of using adaptive knowledge distillation is to transfer attention knowledge from *GGENet-T* to the attention map of *GGENet-S*. We included a convolutional layer between the two models with a kernel size of 1 and a stride of 1 in order to solve the shape disparity between the student's and teacher's attention maps. This allows for an appropriately defined loss function between the teacher and the student. To make sure that the student model accurately replicates the teacher model's attention distribution, the difference between the teacher's attention map and the student's attention map was measured using mean squared error loss. We averaged the three different mean squared error losses of the attention maps between the teacher and student models. Equation (14) provides the hidden representation loss by averaging the loss values of the attention maps.

$$loss_{hidden} = \frac{mse_{attn1} + mse_{attn2} + mse_{attn3}}{3} \tag{14}$$

where $mse_{attn1}$, $mse_{attn2}$, and $mse_{attn3}$ represent the loss values for each attention map between the student and teacher.

We used the dynamic weight as described in 'Adaptive Knowledge Distillation'. The weight of the $hidden_{loss}$ is calculated as given in Eq. (15).

$$weight_{hidden} = max\left(0.25, min\left(\frac{loss_{hidden}}{loss_{hidden} + loss_{CE}}, 0.75\right)\right) \tag{15}$$

where $loss_{CE}$ is the cross entropy loss calculated from the student model. Also, $weight_{hidden}$ is constrained within a specific range, between 0.25 and 0.75, because of the need to balance the model's learning dynamics. This constraint prevents the model from overly relying on certain hidden features, ensuring that the attention is distributed more evenly across the feature maps, which can lead to more stable training and better generalization.

The weight of the cross entropy loss, which is used for student model, is calculated as $weight_{CE} = 1 - weight_{hidden}$.

Total loss is calculated as given in Eq. (16).

$$loss_{total} = weight_{hidden} \times loss_{hidden} + weight_{CE} \times loss_{CE} \qquad (16)$$

where $loss_{hidden}$ represents the loss associated with the hidden features of attention map, and $loss_{CE}$ is the cross-entropy loss calculated from the final output of the student model. The terms $weight_{hidden}$ and $weight_{CE}$ are the corresponding weights applied to these loss components, which control the contribution of each loss term to the total loss.

By distilling the attention knowledge, it is transferred from the teacher model to the student model. This not only transfers the performance of the teacher model to the student model but also allows the student model to be lighter, leading to faster inference. This method ensures performance improvement and results in a faster model.

## EXPERIMENTAL RESULTS

Performance evaluation was conducted for each fold that contains 2,728 test images. The *GGENet-S*, *GGENet-T*, and *GGENet-AKD* models—which represent the teacher, student, and student model with adaptive knowledge distillation, respectively—were put to the testing. Different metrics was used to compare the models.

The accuracy metric, which is used to calculate the percentage of correctly identified examples relative to all instances, is provided by Eq. (17).

$$accuracy = \frac{TP + TN}{TP + FP + TN + FN}. \qquad (17)$$

Equation (18) gives the precision, or the percentage of accurate positive predictions among all positive predictions generated by the model. It is especially crucial in scenarios where false positives have significant consequences since it indicates how accurate the positive predictions are.

$$precision = \frac{TP}{TP + FP}. \qquad (18)$$

Recall is a different metric that indicates the percentage of actual positive cases among all true positive forecasts. It evaluates how successfully the model locates each pertinent instances in the dataset. Equation (19) gives the recall metric.

$$recall = \frac{TP}{TP + FN}. \qquad (19)$$

Equation (20) gives the F1-score, which provides a proper equilibrium between the precision and recall measures. The F1-score, which is the harmonic mean of recall and precision, offers a single metric that combines the two aspects of model performance. When there are uneven class distributions and a balance between recall and precision is needed, it is especially helpful.

**Table 4 Model performance outcomes for each fold on the training set.**

| Fold | Model | Accuracy | Precision | Recall | F1-score | AUC score |
|---|---|---|---|---|---|---|
| 1 | GGENet-T | 1.0 | 1.0 | 1.0 | 1.0 | 1.0 |
| | GGENet-S | 1.0 | 1.0 | 1.0 | 1.0 | 1.0 |
| | GGENet-AKD | 1.0 | 1.0 | 1.0 | 1.0 | 1.0 |
| 2 | GGENet-T | 1.0 | 1.0 | 1.0 | 1.0 | 1.0 |
| | GGENet-S | 0.9980 | 0.9980 | 0.9980 | 0.9980 | 0.9988 |
| | GGENet-AKD | 1.0 | 1.0 | 1.0 | 1.0 | 1.0 |
| 3 | GGENet-T | 1.0 | 1.0 | 1.0 | 1.0 | 1.0 |
| | GGENet-S | 0.9973 | 0.9974 | 0.9973 | 0.9973 | 0.9984 |
| | GGENet-AKD | 1.0 | 1.0 | 1.0 | 1.0 | 1.0 |
| 4 | GGENet-T | 1.0 | 1.0 | 1.0 | 1.0 | 1.0 |
| | GGENet-S | 1.0 | 1.0 | 1.0 | 1.0 | 1.0 |
| | GGENet-AKD | 1.0 | 1.0 | 1.0 | 1.0 | 1.0 |
| 5 | GGENet-T | 1.0 | 1.0 | 1.0 | 1.0 | 1.0 |
| | GGENet-S | 0.9983 | 0.9983 | 0.9983 | 0.9983 | 0.9990 |
| | GGENet-AKD | 1.0 | 1.0 | 1.0 | 1.0 | 1.0 |
| Average | GGENet-T | 1.0 | 1.0 | 1.0 | 1.0 | 1.0 |
| | GGENet-S | 0.9987 | 0.9987 | 0.9987 | 0.9987 | 0.9992 |
| | GGENet-AKD | 1.0 | 1.0 | 1.0 | 1.0 | 1.0 |

$$F1 = 2 \times \frac{precision \times recall}{precision + recall}. \tag{20}$$

We assessed the overall performance of the models using the area under the curve (AUC) score in addition to the metrics that were supplied. The AUC score is obtained using the receiver operating characteristic (ROC) curve, which plots the true positive rate against the false positive rate over a range of threshold values. This statistic is used to assess the model's ability to distinguish between classes; a larger AUC indicates better performance.

The Pytorch framework was used to conduct the training procedure. Every model was trained for a total of 100 epochs using the Adam (*Kingma, 2014*) optimizer, a batch size of 16, and a learning rate of 0.001. The loss function that was employed was cross entropy loss. Before being fed into the models, the input images were scaled down to 224 × 224 pixels. The mean and standard deviation values from the Imagenet (*Deng et al., 2009*) dataset were also used to normalize the images. Better performance was obtained by normalizing the images around zero by subtracting the mean and dividing by the standard deviation for each channel.

## Findings and discussion

The designed models *GGENet-T*, *GGENet-S* and *GGENet-AKD* models were trained and tested separately.

The outcomes from the training set are presented in Table 4.

Both *GGENet-T* and *GGENet-AKD* regularly obtain perfect scores across all metrics and folds in Table 4, demonstrating that they completely fit the training data. In comparison, *GGENet-S* performs worse in folds 2, 3, and 5, with the AUC score slightly below 1.0 and accuracy, precision, recall, and F1-score ranging from 0.9973 to 0.9983. Even with these minor variations, *GGENet-S* continues to function remarkably well. Adaptive knowledge distillation improves the student model's performance on the training data, while the teacher model maintains similar scores to those of the *GGENet-AKD* model.

The validation set findings are demonstrated in Table 5.

In Table 5, *GGENet-AKD* continuously produces competitive and frequently better performance across all folds when compared to the other models. In terms of precision, recall, and F1-score, *GGENet-AKD* performs better than both *GGENet-T* and *GGENet-S*, suggesting a more robust and balanced classification capability, even if *GGENet-T* exhibits greater average accuracy. The average AUC score of 0.9881 for the *GGENet-AKD* model demonstrates strong discriminative power, being higher than those of *GGENet-S* and *GGENet-T*. These findings highlight how adaptive knowledge distillation can enhance the student model's ability to generalize to new data, narrowing the gap with the teacher model while maintaining computational efficiency.

Table 6 shows the outcomes of the testing phase.

In the majority of folds, the *GGENet-AKD* model consistently performs better than both *GGENet-S* and *GGENet-T*, as seen in Table 6. For example, folds 1 through 4 exhibit robust and well-balanced performance, achieving the highest scores across all metrics. The accuracy of fold 2 is noteworthy as it surpasses the other models by a wide margin, reaching 0.9857, highlighting the benefit of adaptive knowledge distillation. Fold 5 is the only exception, where *GGENet-T* slightly outperforms in all metrics.

The most dependable model, with the greatest average scores in all metrics, is *GGENet-AKD* when averaged across all folds. This implies that adaptive knowledge distillation improves the compressed model's predictive performance beyond that of its larger equivalent, in addition to increasing its efficiency. The most reliable option of each of them is *GGENet-AKD* since it provides the best trade-off between model size and classification accuracy.

These performance outcomes clearly show how well the model can distinguish between fresh and rotting fruit. Its ability to differentiate between these two groups is demonstrated by the consistently high accuracy, precision, and recall attained by *GGENet-AKD*.

The confusion matrices for each fold of the *GGENet-AKD* model on the testing set are shown in Fig. 6.

The *GGENet-AKD* model continuously performs better at distinguishing between classes, according to our examination of the AUC scores. This model has the greatest AUC scores across several folds, demonstrating its remarkable ability to rank predictions more accurately.

From fold 1 to fold 4, *GGENet-AKD* consistently outperforms the *GGENet-T* and *GGENet-S* models, achieving the best AUC scores.

Through all folds, *GGENet-AKD* maintains its lead with an average AUC score of 0.9891. This suggests that it performs better than the *GGENet-T* and *GGENet-S* models in

**Table 5 Model performance outcomes for each fold on the validation set.**

| Fold | Model | Accuracy | Precision | Recall | F1-score | AUC score |
|---|---|---|---|---|---|---|
| 1 | GGENet-T | 0.9853 | 0.9854 | 0.9853 | 0.9852 | 0.9912 |
| | GGENet-S | 0.9788 | 0.9789 | 0.9788 | 0.9787 | 0.9873 |
| | GGENet-AKD | 0.9834 | 0.9837 | 0.9834 | 0.9834 | 0.9900 |
| 2 | GGENet-T | 0.9797 | 0.9802 | 0.9797 | 0.9798 | 0.9878 |
| | GGENet-S | 0.9742 | 0.9748 | 0.9742 | 0.9742 | 0.9845 |
| | GGENet-AKD | 0.9788 | 0.9789 | 0.9788 | 0.9788 | 0.9873 |
| 3 | GGENet-T | 0.9751 | 0.9753 | 0.9751 | 0.9752 | 0.9851 |
| | GGENet-S | 0.9705 | 0.9710 | 0.9705 | 0.9705 | 0.9823 |
| | GGENet-AKD | 0.9788 | 0.9789 | 0.9788 | 0.9788 | 0.9873 |
| 4 | GGENet-T | 0.9761 | 0.9766 | 0.9761 | 0.9761 | 0.9856 |
| | GGENet-S | 0.9751 | 0.9755 | 0.9751 | 0.9750 | 0.9851 |
| | GGENet-AKD | 0.9770 | 0.9774 | 0.9770 | 0.9769 | 0.9862 |
| 5 | GGENet-T | 0.9816 | 0.9818 | 0.9816 | 0.9815 | 0.9890 |
| | GGENet-S | 0.9724 | 0.9725 | 0.9724 | 0.9724 | 0.9834 |
| | GGENet-AKD | 0.9834 | 0.9836 | 0.9834 | 0.9834 | 0.9901 |
| Average | GGENet-T | 0.9795 | 0.9798 | 0.9795 | 0.9795 | 0.9878 |
| | GGENet-S | 0.9742 | 0.9745 | 0.9742 | 0.9741 | 0.9845 |
| | GGENet-AKD | 0.9782 | 0.9805 | 0.9802 | 0.9802 | 0.9881 |

**Table 6 Outcomes of the models' performance on the testing set at every fold.**

| Fold | Model | Accuracy | Precision | Recall | F1-score | AUC score |
|---|---|---|---|---|---|---|
| 1 | GGENet-T | 0.9780 | 0.9779 | 0.9780 | 0.9779 | 0.9868 |
| | GGENet-S | 0.9798 | 0.9798 | 0.9798 | 0.9797 | 0.9879 |
| | GGENet-AKD | 0.9805 | 0.9806 | 0.9805 | 0.9805 | 0.9883 |
| 2 | GGENet-T | 0.9787 | 0.9790 | 0.9787 | 0.9787 | 0.9872 |
| | GGENet-S | 0.9724 | 0.9728 | 0.9724 | 0.9723 | 0.9835 |
| | GGENet-AKD | 0.9857 | 0.9858 | 0.9857 | 0.9856 | 0.9914 |
| 3 | GGENet-T | 0.9809 | 0.9809 | 0.9809 | 0.9806 | 0.9885 |
| | GGENet-S | 0.9743 | 0.9745 | 0.9743 | 0.9743 | 0.9846 |
| | GGENet-AKD | 0.9853 | 0.9853 | 0.9853 | 0.9853 | 0.9912 |
| 4 | GGENet-T | 0.9783 | 0.9785 | 0.9783 | 0.9783 | 0.9870 |
| | GGENet-S | 0.9776 | 0.9777 | 0.9776 | 0.9775 | 0.9865 |
| | GGENet-AKD | 0.9812 | 0.9815 | 0.9812 | 0.9812 | 0.9887 |
| 5 | GGENet-T | 0.9809 | 0.9810 | 0.9809 | 0.9808 | 0.9885 |
| | GGENet-S | 0.9702 | 0.9704 | 0.9702 | 0.9701 | 0.9821 |
| | GGENet-AKD | 0.9764 | 0.9766 | 0.9764 | 0.9764 | 0.9859 |
| Average | GGENet-T | 0.9793 | 0.9794 | 0.9793 | 0.9791 | 0.9876 |
| | GGENet-S | 0.9748 | 0.9750 | 0.9748 | 0.9747 | 0.9849 |
| | GGENet-AKD | 0.9818 | 0.9820 | 0.9818 | 0.9818 | 0.9891 |

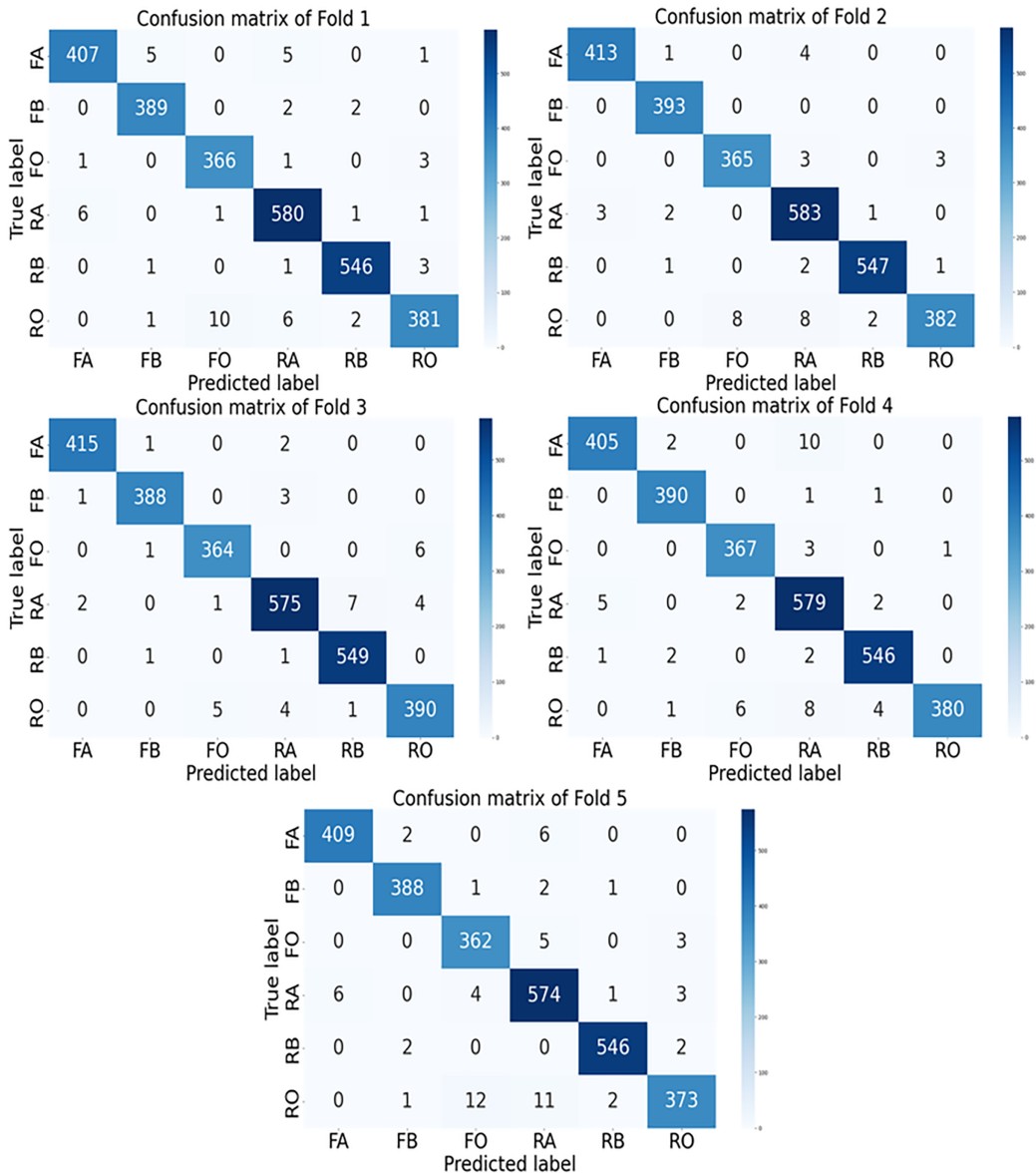

**Figure 6 The confusion matrix of *GGENet-AKD* in each fold on the testing set.** In the figures, FA, FB, FO, RA, RB, and RO represent fresh apple, fresh banana, fresh orange, rotten apple, rotten banana, and rotten orange, respectively.

classifying the freshness of fruit. Even when class distinctions are subtle, the model's superior ability to discriminate between fresh and rotting fruit is reflected in the higher AUC score.

A performance comparison of the suggested model and the research in the literature is given in Table 7. Because they characterized the fruits—banana, apple, and orange—as fresh or rotting and used the same dataset as ours, a comparison with the research in the table was conducted.

| Article | Method | Accuracy | Precision | Recall | F1-score |
|---|---|---|---|---|---|
| *Rohit Mamidi et al. (2022)* | Xception | 0.9725 | – | – | – |
| *Mishra & Singh (2024)* | VGG16 | 0.9750 | 0.9780 | 0.9690 | 0.9730 |
| *Srinivas & Yadiah (2022)* | CNN | 0.9700 | – | – | – |
| *Palakodati et al. (2020)* | CNN | 0.9782 | – | – | – |
| *Bindu et al. (2021)* | CNN | 0.9488 | – | – | – |
| *Sangam et al. (2023)* | CNN | 0.9700 | 0.9700 | 0.9700 | 0.9700 |
| *Nerella et al. (2023)* | Inceptionv3 | 0.9710 | – | – | – |
| Proposed model | *GGENet-AKD* | 0.9818 | 0.9820 | 0.9818 | 0.9818 |

**Table 7** comparison between the suggested model and the studies obtained from the literature.

Out of all the models examined, the suggested *GGENet-AKD* model performs the most robustly and evenly. It has a significant capacity to generalize effectively and perform classification tasks reliably, reducing false positives and false negatives, with the highest accuracy and consistent precision, recall, and F1-score values. It is especially well-suited for real-world applications where both sensitivity and specificity are crucial due to its balanced performance.

According to *Mishra & Singh (2024)*, VGG16 is another model that performs well among the comparisons. Its high accuracy and recall indicate that it can handle both accurate classifications and error reduction. Its lower accuracy and F1-score suggest that it might not be as consistently reliable as the proposed approach.

It is challenging to completely evaluate the dependability of models such as Inceptionv3 (*Nerella et al., 2023*) and Xception (*Rohit Mamidi et al., 2022*) in real-world situations since they have competitive accuracy scores without providing comprehensive breakdowns of other evaluation metrics. A number of convolutional neural network based models, including those from *Sangam et al. (2023)*, *Srinivas & Yadiah (2022)*, and *Palakodati et al. (2020)*, also report comparatively high accuracy, but they do not provide thorough analyses across key performance indicators.

The suggested *GGENet-AKD* strategy offers both higher accuracy and balanced, comprehensive metric reporting, making it a more reliable solution for classification tasks in this area, even if some current models perform well in terms of accuracy.

These comparison results further support the effectiveness of *GGENet-AKD* in the binary task of identifying fresh or rotting fruit. Its usefulness for real-world inspection tasks, where precise classification is crucial for cutting waste and guaranteeing quality control, is highlighted by its capacity to discern fruit freshness.

## Computation time

In this study, a batch size of 16 and an input image size of 224 were used during testing.

The entire number of parameters in the models is listed in Table 8.

Table 9 provides the elapsed time during testing, which varies based on the number of model parameters.

**Table 8 Number of parameters of the models.**

|  | GGENet-T | GGENet-S | GGENet-AKD |
|---|---|---|---|
| Parameters | 7,190,092 | 2,379,388 | 2,379,388 |

**Table 9 Elapsed time during testing.**

| Fold | Model | Elapsed time (s) |
|---|---|---|
| 1 | GGENet-T | 12.22 |
|  | GGENet-S | 10.80 |
|  | GGENet-AKD | 11.43 |
| 2 | GGENet-T | 14.11 |
|  | GGENet-S | 10.76 |
|  | GGENet-AKD | 10.92 |
| 3 | GGENet-T | 14.05 |
|  | GGENet-S | 10.71 |
|  | GGENet-AKD | 10.72 |
| 4 | GGENet-T | 14.45 |
|  | GGENet-S | 10.80 |
|  | GGENet-AKD | 10.87 |
| 5 | GGENet-T | 14.18 |
|  | GGENet-S | 10.98 |
|  | GGENet-AKD | 10.82 |
| Average | GGENet-T | 13.80 |
|  | GGENet-S | 10.81 |
|  | GGENet-AKD | 10.95 |

It is evident by looking at the table that each model's elapsed time differs between the folds.

With delays ranging from roughly 1.4 to 3.7 s, *GGENet-T* consistently performs functions more slowly than *GGENet-S* across all five folds. *GGENet-AKD*, on the other hand, displays processing speeds that are extremely similar to *GGENet-S*, with variations typically less than 0.2 s—in some cases, even significantly faster. These findings demonstrate that although the larger architecture of *GGENet-T* results in longer computation times, the distillation procedure in *GGENet-AKD* retains performance and efficiency that are on equal with those of the original *GGENet-S*.

On average, *GGENet-T* consistently takes longer to complete tasks than both *GGENet-S* and *GGENet-AKD* across all folds. Specifically, *GGENet-T* is approximately 3 s slower than the other two models on average. *GGENet-T* has a longer elapsed time than other models since it demands more processing power due to its larger number of parameters. This is to be expected because *GGENet-T* is designed to handle more complex tasks, although at the cost of longer processing times. Both *GGENet-S* and *GGENet-AKD* share the same number of parameters. The minimal difference in their elapsed times indicates that the adaptive

knowledge distillation process did not significantly affect the computational efficiency of *GGENet-AKD*, keeping it almost as fast as *GGENet-S*.

The attention knowledge that is moved from the *GGENet-T* model to the *GGENet-S* model increases performance without increasing elapsed time, according to an analysis of performance in Table 6 and elapsed time in Table 9. This suggests that adaptive knowledge distillation improves the model's precision and efficiency in identifying rotting or fresh fruit. This work thus shows how adaptive knowledge distillation contributes to more efficient and scalable fruit freshness classification in addition to improving model performance and time efficiency.

In the context of practical applications, especially those involving deployment on low-power edge devices, the performance advantages made possible by the suggested adaptive knowledge distillation approach may seem insignificant. The comparatively minor improvement results from the conventionally explained student model's already high baseline performance, which naturally reduces the opportunity for improvement. Furthermore, it's crucial to remember that as the performance ceiling gets closer, improving model accuracy gets harder and harder. For example, increasing accuracy from 0.80 to 0.90 is usually easier to do, whereas increasing accuracy from 0.97 to 0.98 or 0.99 necessitates more complex optimization and produces similarly important effects. The suggested approach successfully transfers attention knowledge from the teacher model to the student, allowing for competitive performance with reduced computing needs, despite the small numerical difference.

## CONCLUSION

We developed a new deep learning model in this work, namely *GGENet*, specifically designed to discriminate between fresh and rotting fruits. The necessity for a quick and high-performing deep learning model for classifying fruit freshness in practical settings is the main driving force for this investigation. There are two versions of our model: *GGENet-S*, which is the student, and *GGENet-T*, which is the teacher. We managed to successfully convey attention knowledge from the teacher to the student model by using adaptive knowledge distillation.

The GRN-GELU Conv module is a new module that we introduced. To improve feature extraction, this module combines Gaussian error linear unit activation, global response normalization, and a residual structure. Additionally, spatial and channel attention mechanisms were successfully integrated into the network. The proposed model demonstrated better performance than the large model with an accuracy of 0.9818 and is approximately 3 s faster, with an elapsed time of 10.95 s. These results demonstrate how the developed model benefits from efficient knowledge transfer and optimization, achieving high performance while maintaining computational efficiency.

The creation of the new GRN-GELU Conv module and the incorporation of adaptive knowledge distillation specifically suited for fruit freshness classification are the study's main contributions. The model's capacity to extract significant features and sustain high classification accuracy with shorter inference times is directly improved by these advancements. The suggested *GGENet-S* model outperforms its larger equivalents while

running faster, striking a convincing balance between performance and efficiency in contrast to conventional models. Thus, this work advances the discipline by providing a useful framework for adaptive knowledge distillation that is optimized for attention transfer, in addition to offering a workable solution to an actual agricultural problem.

While the results are encouraging, our method has limitations. First, because both teacher and student model inference are required during training, the adaptive knowledge distillation process adds computing overhead and training complexity. Second, the model's performance might be influenced in real-world applications and environments. Therefore, the developed model should be tested in practical settings to assess its effectiveness and robustness in diverse conditions. Lastly, the quality and compatibility of teacher-student pairings have a significant impact on the efficacy of attention transfer and adaptive knowledge distillation; a poorly selected teacher may misdirect the student, lowering performance improvements.

In future studies, the developed model will be integrated into Internet of Things devices, allowing it to be tested in real-world environments. This integration will help evaluate its performance and robustness in practical settings, providing insights into how well it operates under varied conditions and with real-time data. The model's capacity to generalize will be further improved by enlarging the dataset to encompass a wider variety of fruit varieties, lighting scenarios, and stages of deterioration.

### Funding
The authors received no funding for this work.

### Competing Interests
The authors declare that they have no competing interests.

### Author Contributions
- Semih Demirel conceived and designed the experiments, performed the experiments, analyzed the data, performed the computation work, prepared figures and/or tables, authored or reviewed drafts of the article, and approved the final draft.
- Oktay Yıldız performed the experiments, analyzed the data, prepared figures and/or tables, authored or reviewed drafts of the article, and approved the final draft.

### Data Availability
The source code is available at Zenodo:

Semih Demirel. (2025). semihhdemirel/GGENet: GGENet_Main_Release (GGENet_Main_Release). Zenodo. https://doi.org/10.5281/zenodo.15727051.

The dataset and checkpoints are available at Zenodo:

DEMİREL, S. (2025). Classification of fruits as fresh and rotten [Data set]. Zenodo. https://doi.org/10.5281/zenodo.15719207.

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
