# Peer review of "Enhancing fruit freshness classification with adaptive knowledge distillation and global response normalization in convolutional networks"

_PeerJ Computer Science, doi:10.7717/peerj-cs.3198_

## Round 0.1 · original submission · Major Revisions

Reviewer 1 ·

Basic reporting

One paragraph is needed to be added to the introduction to show the main contribution and how does it differ from the previous work.
The author should add a paragraph at the end of the introduction section to describe the paper structure and sections.
Authors need to confirm that all acronyms are defined before being used for first time. Authors need to confirm that all mathematical notations are defined when being used for first time.
Improve the readability of the manuscript in terms of typos mistakes and errors. There are some grammatical errors and awkward phrasings (suggested proofreading the manuscript after addressing all comments to avoid any typo, grammatical, and lingual mistakes and errors).
Some references need to be updated …

Experimental design

I miss a section that outlines the limitations of your approach and possibilities of extension. Are there any disadvantages or limits of your method?

Validity of the findings

The conclusion needs improvements towards major claimed contribution.
Write some future directions in the conclusion section.

·

Basic reporting

1. When presenting the results, the author tends to mention every single result already shown in the accompanying figure/table and this leads the article to be needlessly long. It would be better if the authors only discussed the most significant results and explain the contextual meaning behind them.

2. The names of the models used for this study are too long and too similar. For reporting purposed, it is recommended to choose names that are easily differentiable. Also, i believe the model names should be italicised so as to not appear as normal text.

3. In the first page, there should be a segway between the introduction of the importance of agriculture and the computer science sections for a better transition flow for the reader.

4. Page 3, paragraph 1 should be broken down to smaller paragraphs.

5. The reporting on the fruit freshness classification disappears after the description of the dataset, as the author focuses wholly on the model performance outside of the context of describing fruit freshness. Consider presenting the results within the context of fruit classification as indicated in the manuscript title.

Experimental design

1. Please clarify why the authors chose to test the concept of knowledge distillation in this fruit freshness classification task and not other classification tasks. \

2. Since it is the most novel part of the manuscript, i think more explanations should be given for the knowledge distillation technique in section 3.4

Validity of the findings

1. The concept of AI knowledge distillation is an interesting one, but the differences between training the smaller model conventionally and with this new technique seems to be minimal (<0.01 for accuracy and only <0.15s computation time). The author summarised that this technique is an improvement from the conventional technique but the results are not convincing. Perhaps the author can discuss more on why the results are the way they are and propose how to improve the performance of the smaller model with this proposed technique.

2. In Table 5, are all the models being compared to here using the same dataset used by the author ? Otherwise, the comparisons does not hold much weight since it is not "apple to apple".

3. Why are there no results of the training process ?

Additional comments

1. The citation style here does not follow the journal format in that the reference names are not enclosed within bracket and therefore looks messy.

2. Overall format needs to rechecked. For example, links to the source code should not be included in the abstract.

---

## Round 0.2 · accepted · Accept

Many thanks to the authors for their efforts to improve the article. Based on the comments of the reviewer, I believe this version addressed the issues successfully. It can be accepted.

Reviewer 1 ·

Basic reporting

In the revised version of the manuscript, the authors met all the requirements and comments given in the previous review, so I recommend this paper for publishing.

Experimental design

In the revised version of the manuscript, the authors met all the requirements and comments given in the previous review, so I recommend this paper for publishing.

Validity of the findings

In the revised version of the manuscript, the authors met all the requirements and comments given in the previous review, so I recommend this paper for publishing.

Additional comments

In the revised version of the manuscript, the authors met all the requirements and comments given in the previous review, so I recommend this paper for publishing.